# Don't Lag, RAG: Training-Free Adversarial Detection Using RAG

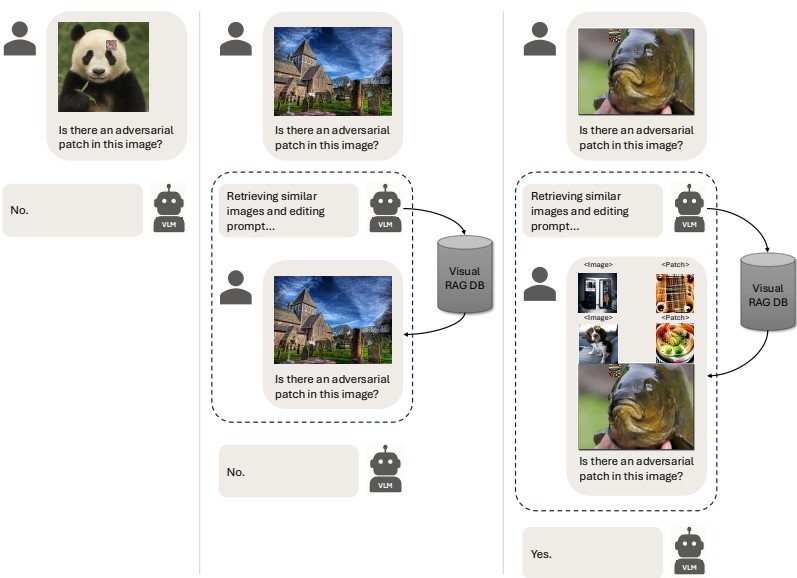

Figure 1: Illustration of three different settings for detecting adversarial patches. (Left) The zero-shot baseline, in which the model is directly prompted to determine if the image is adversarial but incorrectly concludes it is benign. (Center) Our VRAG-based approach on a benign image; as the database does not contain benign exemplars, no relevant references are retrieved. Consequently, the classification relies solely on the prompt content and remains accurate. (Right) Our VRAG-based approach on an adversarial image, which leverages relevant references from the database to enhance the prompt, ultimately yielding a correct detection of the adversarial patch.

## Abstract

Adversarial patch attacks pose a major threat to vision systems by embedding localized perturbations that mislead deep models. Traditional defense methods often require retraining or fine-tuning, making them impractical for real-world deployment. We propose a training-free Visual Retrieval-Augmented Generation (VRAG) framework that integrates Vision-Language Models (VLMs) for adversarial patch detection. By retrieving visually similar patches and images that resemble stored attacks in a continuously expanding database, VRAG performs generative reasoning to identify diverse attack types-all without additional training or fine-tuning. We extensively evaluate open-source large-scale VLMs-including Qwen-VL-Plus, Qwen2.5-VL-72B, and UI-TARS-72B-DPO-alongside Gemini-2.0, a closed-source model. Notably, the open-source UI-TARS-72B-DPO model achieves up to 95% classification accuracy, setting a new state-of-the-art for open-source adversarial patch detection. Gemini-2.0 attains the highest overall accuracy, 98%, but remains closed-source. Experimental results demonstrate VRAG's effectiveness in identifying a variety of adversarial patches with minimal human annotation, paving the way for robust, practical defenses against evolving adversarial patch attacks.

## 1 INTRODUCTION

Deep learning models, particularly convolutional neural networks (CNNs) Krizhevsky et al. (2012); He et al. (2016); Simonyan & Zisserman (2014) and vision transformers (ViTs) Dosovitskiy et al. (2020), have demonstrated remarkable success in computer vision tasks such as object detection Girshick (2015); Ren et al. (2016); Redmon et al. (2016), image classification Krizhevsky et al. (2012); Dosovitskiy et al. (2020), and segmentation Long et al. (2015); Ronneberger et al. (2015). However, despite advances, these models remain highly vulnerable to adversarial attacks Szegedy (2013); Lapid & Sipper (2023); Lapid et al. (2022); Alter et al. (2025); Goodfellow et al. (2014); Madry et al. (2017); Tamam et al. (2023); Lapid et al. (2024b), where small perturbations or carefully crafted patches manipulate predictions.

Adversarial patch attacks Brown et al. (2017); Lapid et al. (2024c); Liu et al. (2018); Wei et al. (2023) introduce localized perturbations that persist across different transformations, making them significantly more challenging to mitigate using conventional defense mechanisms Wei et al. (2022). Unlike traditional adversarial perturbations that introduce subtle noise across an image, adversarial patches are structured, high-magnitude perturbations, which are often physically realizable (Lee & Kolter, 2019; Hu et al., 2021). These patches can be printed, placed in real-world environments, and still cause misclassification or mislocalization in deployed deep learning models. Their adversarial effect remains robust under different lighting conditions, transformations, and occlusions, allowing them to be successfully deployed in real world scenarios Liu et al. (2024); Deng et al. (2023). Furthermore, retraining-based defenses require extensive, labeled adversarial data, which is expensive to obtain and generalizes poorly to novel attack strategies Wei et al. (2022).

Traditional adversarial detection methods typically fall into one of three categories, (1) supervised learning-based defenses, (2) unsupervised defenses and (3) adversarial training. Supervised learning-based defenses Pinhasov et al. (2024); Papernot et al. (2016) use deep learning classifiers trained on labeled adversarial and non-adversarial samples. These methods are data-dependent and do not adapt well to adversarial attacks outside the training distribution. Unsupervised defenses Xu et al. (2018); Papernot & McDaniel (2018); Sotgiu et al. (2020); Mizrahi et al. (2025), typically rely on analyzing the intrinsic structure or distribution of unlabeled data to detect anomalous inputs. For example, Feature Squeezing Xu et al. (2018) reduces input dimensionality (e.g., through bit-depth reduction or smoothing) to reveal suspicious high-frequency artifacts; Papernot & McDaniel (2018) use deep generative models to flag inputs with high reconstruction error as potential adversarial samples. Although these methods can detect novel or previously unseen attack strategies without relying on adversarial labels, they often require carefully chosen hyperparameters and remain vulnerable to adaptive attacks that mimic the statistics of benign inputs. In contrast to the supervised detection methods, which *separately classify* inputs as adversarial or benign, *adversarial training* Madry et al. (2017); Lapid et al. (2024a) augments the training data with adversarial examples to directly improve model robustness. Rather than solely learning to detect adversarial inputs, this approach modifies the model parameters and decision boundaries to make correct classification more likely under attack. However, adversarial training is computationally expensive and risks overfitting to specific attack types, leading to weaker defenses against unseen attacks Liang et al. (2024).

In this paper, we introduce a retrieval-augmented adversarial patch detection framework that dynamically adapts to evolving threats without necessitating retraining. The method integrates visual retrieval-augmented generation (VRAG) with a vision-language model (VLM) for context-aware detection. As illustrated in 1, visually similar patches are retrieved from a precomputed database using semantic embeddings from grid-based image regions, and structured natural language prompts guide the VLM to classify suspicious patches.

This paper makes the following contributions:

1. A training-free retrieval-based pipeline that dynamically matches adversarial patches against a precomputed (and expandable) database.

2. The integration of existing VLMs with generative reasoning for context-aware patch detection through structured prompts.

3. A comprehensive evaluation demonstrating robust detection across diverse adversarial patch scenarios, all without additional training or fine-tuning.

Experimental results confirm that our retrieval-augmented detection approach not only outperforms traditional classifiers, but also achieves state-of-the-art detection across a variety of threat scenarios. This method offers higher accuracy and reduces dependence on labeled adversarial datasets, underscoring the practicality of incorporating retrieval-based strategies alongside generative reasoning to develop scalable, adaptable defenses for real-world security applications Kazoom et al. (2024).

## 2 RELATED WORK

Adversarial attacks exploit neural network vulnerabilities through carefully crafted perturbations. Early works focused on small, imperceptible $\ell_p$-bounded perturbations such as FGSM Goodfellow et al. (2014) and PGD Madry et al. (2017). In contrast, *adversarial patch attacks* apply localized, high-magnitude changes that remain effective under transformations and pose a threat in real-world scenarios Hwang et al. (2023); Liu et al. (2024); Deng et al. (2023).

Defenses fall into *reactive* and *proactive* categories. Reactive methods like JPEG compression Dziugaite et al. (2016) and spatial smoothing Xu et al. (2017) attempt to remove adversarial patterns at inference time but struggle against adaptive attacks. Diffusion-based methods, such as DIFFender Kang et al. (2024) and purification models Lin et al. (2023), leverage generative models to restore clean content but are often computationally intensive.

Another line of work focuses on *patch localization and segmentation*, e.g., SAC Liu et al. (2022), which detects and removes patches using segmentation networks. These approaches are limited by their reliance on training and struggle with irregular or camouflaged patches. PatchCleanser Xiang et al. (2022) offers certifiable robustness but assumes geometrically simple patches.

Proactive defenses like adversarial training Wei et al. (2022) aim to increase robustness through exposure to adversarial examples. While effective against known attacks, they generalize poorly and are resource-intensive.

We propose a retrieval-augmented framework that detects a wide range of patch types-including irregular and naturalistic ones (7) -without degrading input quality or relying on segmentation or geometric assumptions. Our method leverages a diverse patch database and vision-language reasoning to dynamically adapt to unseen attacks.

## 3 PRELIMINARIES

We briefly review core paradigms relevant to our defense framework: vision-language foundation models, zero- and few-shot learning, adversarial attacks and defenses, and RAG.

### 3.1 VISION-LANGUAGE FOUNDATION MODELS AND ZERO- AND FEW-SHOT LEARNING

Foundation models leverage large-scale transformer (Dosovitskiy et al., 2020) architectures and self-attention (Vaswani et al., 2017) to learn general-purpose representations from massive image-text data. A typical *VLM* consists of two encoders, $f_\theta$ for images $I$ and $g_\phi$ for text $T$, projecting them into a shared embedding space:

$$E_I = f_\theta(I), \quad E_T = g_\phi(T), \quad S(I,T) = \frac{E_I \cdot E_T}{\|E_I\|\|E_T\|}. \tag{1}$$

Models like CLIP Radford et al. (2021) and Flamingo Alayrac et al. (2022) align image-text pairs via contrastive objectives, enabling flexible *zero-shot* capabilities:

$$g(I, Q) \to A, \tag{2}$$

where $Q$ is a textual query and $A$ is the inferred label without explicit task-specific training. *Few-shot learning* refines zero-shot by supplying a small support set $\{(I_1, y_1), \ldots, (I_k, y_k)\}$:

$$g\big(I, Q \mid \{(I_i, y_i)\}_{i=1}^k\big) \to A, \tag{3}$$

allowing adaptation to novel tasks with limited labeled data.

## 3.2 ADVERSARIAL ATTACKS AND DEFENSE STRATEGIES

**Adversarial Attacks.** Formally, an adversary seeks a perturbation $\delta$ subject to $\|\delta\|_p \leq \epsilon$ that maximizes a loss function $\ell$ for a model $f_\theta$ with true label $y$:

$$\delta^* = \arg \max_{\|\delta\|_p \leq \epsilon} \ell\big(f_\theta(I + \delta), y\big). \tag{4}$$

Patch-based attacks instead replace a localized region using a binary mask $M \in \{0, 1\}^{H \times W}$:

$$I' = I \odot (1 - M) + P \odot M, \tag{5}$$

where $P$ is a high-magnitude patch. Since the threat model is specified solely by the support of $M$, *no $\epsilon$–norm constraint is imposed on the pixel values inside the patch*; the perturbation can therefore have unbounded $\ell_p$ magnitude within $M$ while remaining spatially confined Hwang et al. (2023); Liu et al. (2024); Deng et al. (2023).

**Preprocessing and Detection.** A common defense strategy is to apply a transformation $g(\cdot)$ to $I'$, yielding $g(I')$, with the goal of suppressing adversarial noise (e.g., blurring, smoothing Kim et al. (2022)). Detection can be formulated by a function $D\big(g(I')\big) \in \{0, 1\}$ that flags anomalous inputs based on statistical or uncertainty-based criteria Chua et al. (2022).

**Generative Reconstruction.** Diffusion-based defenses Kang et al. (2024) iteratively denoise adversarial inputs by reversing a noisy forward process:

$$x_t = \sqrt{\alpha_t}\, x_{t-1} + \sqrt{1 - \alpha_t}\, \epsilon_t, \quad \epsilon_t \sim \mathcal{N}(0, I), \tag{6}$$

often guided by patch localization Liu et al. (2022). Although effective, these approaches can falter against unseen attacks or large patch perturbations, making robust generalization challenging in practice.

## 3.3 RETRIEVAL-AUGMENTED GENERATION

Retrieval-Augmented Generation (RAG) Lewis et al. (2020) integrates external knowledge into a generative model to improve both its generative capacity and semantic coherence. Formally, given a query $Q$, the model retrieves the top-$k$ most relevant documents or embeddings $R_k$ from a database $\mathcal{D}$:

$$R_k = \arg \max_{R_i \in \mathcal{D}} S(Q, R_i), \tag{7}$$

where $S(\cdot, \cdot)$ is a similarity function. The query $Q$ is then combined with $R_k$ within a generative function:

$$A = G(Q, R_k). \tag{8}$$

In our approach, this retrieval phase facilitates access to known adversarial patches, thereby enabling a more robust generative reasoning process. By incorporating historical data on diverse attack patterns, RAG-based defenses can dynamically adapt to novel threats while sustaining high efficacy against existing adversaries.

## 4 METHODOLOGY

This section details our VRAG-based approach for adversarial patch detection using a vision-language model. We describe the construction of a comprehensive adversarial patch database (§4.1), and then present our end-to-end detection pipeline (§4.2). We discuss how the framework generalizes to diverse patch shapes in §A.6. To enable scalability, we parallelize patch embedding and augmentation-see Appendix A.1 for runtime benchmarks across varying numbers of workers.

## 4.1 DATABASE CREATION

To handle a wide variety of adversarial patch attacks, we build a large-scale database of patched images and their corresponding patch embeddings. We aggregate patches generated by SAC Liu et al. (2022), BBNP Lapid et al. (2024c), and standard adversarial patch attacks Brown et al. (2017), placing each patch onto diverse natural images at random positions and scales. This process, summarized in 1, ensures that the database spans different patch configurations and visual contexts.

---

**Algorithm 1** Adversarial Patch Database Creation with Positional Augmentation

---

1: **Input:** Set of patches $\{P_i\}_{i=1}^m$, set of natural images $\{I_j\}_{j=1}^q$, embedding model $f$, grid size $n \times n$, number of placement variations $A$
2: **Output:** Database $\mathcal{D}$
3: Initialize $\mathcal{D} \leftarrow \emptyset$
4: **for** $i = 1$ **to** $m$ **do**
5:     $E_{P_i} \leftarrow f(P_i)$
6:     Store $(P_i, E_{P_i})$ in $\mathcal{D}$
7:     **for** $j = 1$ **to** $q$ **do**
8:         **for** $a = 1$ **to** $A$ **do**
9:             Randomly select position $(x_a, y_a)$ in image $I_j$
10:            Apply patch $P_i$ at $(x_a, y_a)$ to obtain $I_j^{(a)}$
11:            Divide $I_j^{(a)}$ into grid cells $\{C_{j,k}^{(a)}\}_{k=1}^{n^2}$
12:            **for** $k = 1$ **to** $n^2$ **do**
13:               Compute $E_{j,k}^{(a)} = f(C_{j,k}^{(a)})$
14:               **if** $C_{j,k}^{(a)}$ overlaps $P_i$ **then**
15:                  Store $(C_{j,k}^{(a)}, E_{j,k}^{(a)})$ in $\mathcal{D}$
16:               **end if**
17:            **end for**
18:         **end for**
19:     **end for**
20: **end for**
21: **return** $\mathcal{D}$

---

Concretely, each patched image is subdivided into an $n \times n$ grid, yielding localized regions $\{C_1, \ldots, C_{n^2}\}$ that spatially partition the image. For each region $C_i$, we compute a dense visual embedding using a pre-trained vision encoder $f(\cdot)$:

$$E_{C_i} = f(C_i),$$

which captures high-level semantic and structural features of the corresponding image patch. In parallel, we encode each adversarial patch $P_j$ into its own latent representation $E_{P_j} = f(P_j)$ to ensure embeddings are in the same feature space. These patch embeddings act as *keys*, while the embeddings of overlapping regions serve as their corresponding *values* in a key-value database. This design enables efficient and scalable nearest-neighbor retrieval at inference time, allowing the system to match visual evidence in test images with known adversarial patterns from the database.

### 4.2 VRAG-BASED DETECTION PIPELINE

**System Overview.** Our detection system (illustrated in 2) identifies adversarial patches in a query image by leveraging the patch database as retrieval context for a vision-language model. The process involves four main steps:

1. **Image Preprocessing:** Divide the input image $I$ into an $n \times n$ grid of regions $\{C_1, \ldots, C_{n^2}\}$ to enable localized inspection of each part of the image.

2. **Feature Extraction:** Encode each region $C_i$ into an embedding $E_i = f(C_i)$ using a pre-trained vision encoder (e.g., CLIP). These embeddings capture high-level semantic features.

3. **Retrieval Step:** For each $E_i$, perform a nearest-neighbor search in the patch database $\mathcal{D}$. Retrieve the top-$k$ most similar patch embeddings to form a context set $\mathcal{R}_i = \text{Top-}k(\{d(E_i, E_{P_j})\})$. Appendix A.3 presents an ablation study comparing cosine similarity with alternative distance metrics for this retrieval step.

4. **Generative Reasoning with a VLM:** Combine each region $C_i$ with its retrieved examples $\mathcal{R}_i$ and short textual cues to construct a multimodal prompt. This prompt is passed to a vision-language model $g(\cdot)$ to answer:

$$g(\mathcal{C}_i) \rightarrow \text{``Does this region contain an adversarial patch?''}$$

---

**Algorithm 2** Adversarial Patch Detection via VRAG

---

1: **Input:** Image $I$, VLM $\mathcal{V}$, Database $\mathcal{D}$, Embedding $f$, threshold $\tau$, top-$m$ patches, top-$k$ images

2: **Output:** Decision $\in \{\texttt{Attacked}, \texttt{Not Attacked}\}$

3: Divide $I$ into grid cells $\{C_i\}_{i=1}^{n^2}$; compute embeddings $E_i = f(C_i)$.

4: **for** each $E_i$ **do**

5:    Compute $S_i = \max_{E_d \in \mathcal{D}} \frac{E_i \cdot E_d}{\|E_i\|\|E_d\|}$.

6: **end for**

7: Select candidates $\mathcal{C} = \{C_i \mid S_i \geq \tau\}$; choose top-$m$ patches.

8: Retrieve top-$k$ similar attacked images from $\mathcal{D}$.

9: Build context $\mathcal{T}$ with the selected patches and images.

10: Query VLM: $R = \mathcal{V}(\mathcal{T}, I)$.

11: **return** $R$.

---

We summarize the overall detection procedure in 2.

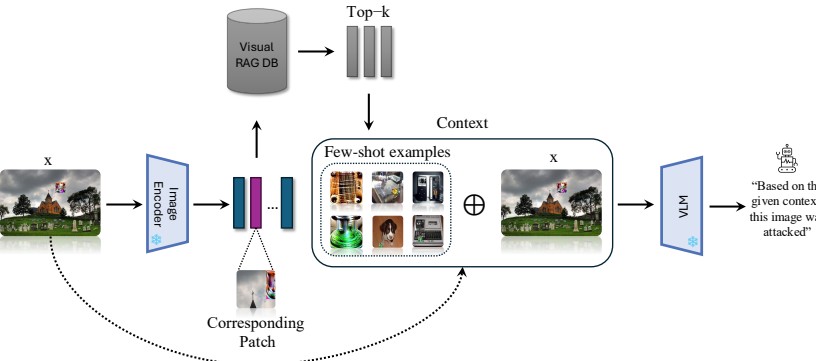

Figure 2: Overview of our VRAG framework for adversarial patch detection. Given a query image, we extract grid-based embeddings and retrieve the top-$k$ visually similar adversarial patches from our database. These patches and their associated attacked images form a few-shot context for a vision-language model that decides whether the query contains an adversarial patch.

**Decision Mechanism (Zero-Shot and Few-Shot).** After retrieving similar patches and attacked images, the VLM is prompted to judge the query image under zero-shot or few-shot conditions:

- *Zero-Shot Detection:* The model relies on pre-trained knowledge and textual prompts to classify each region $C_i$ as adversarial or benign, without additional fine-tuning.

- *Few-Shot Adaptation:* A small, labeled set of adversarial examples, denoted as $\{A_i\}$, along with their corresponding patches $\{P_i\}$, is incorporated into the retrieved context to refine the model's decision-making process. This integration enhances the model's robustness to previously unseen attacks by explicitly exposing the VLM to representative instances of patch-induced behavior.

A sample query prompt for the VLM might be:

```
``Here are examples of adversarial patches:  [Patch 1],
[Patch 2].  Here are images that contain these patches:
[Image 1], [Image 2].  Based on this context, does the
following image contain an adversarial patch?  Answer 'yes'
or 'no'.''
```

The model's answer is then used to decide whether the image is `Attacked` or `Not Attacked`.

**Optimal Threshold Selection.** We determine the optimal threshold based on ROC-AUC analysis of cosine similarity scores computed from embedding vectors. Specifically, the optimal cosine similarity threshold identified was $0.77$, providing the best trade-off between sensitivity and specificity. We observed that for thresholds approaching $1.0$, the similarity criterion becomes overly permissive, resulting in nearly every image retrieving similar images, thereby substantially increasing the false-positive rate.

## 5 EXPERIMENTAL EVALUATION

We conduct extensive experiments to assess the robustness and efficiency of our adversarial patch detection framework across diverse datasets, models, attack types, and defenses, simulating realistic deployment scenarios.

**Vision Language Models.** For generative reasoning, we use several VLMs $g(\cdot)$, including *Qwen-VL-Plus* Cloud (2023), *Qwen2.5-VL-Instruct* Cloud (2024), *UI-TARS-72B-DPO* Research (2024), and *Gemini* DeepMind (2024). These were chosen for their strong multimodal reasoning in zero- and few-shot settings. While *Gemini 2.0* yields the highest accuracy, it is proprietary. *UI-TARS-72B-DPO*, meanwhile, offers competitive performance and sets a strong benchmark among open-source models.

**Classification Models.** To evaluate the impact of adversarial patches across diverse architectures, we consider four representative image classification models: (1) *ResNet-50* He et al. (2016), (2) *ResNeXt-50* Xie et al. (2017), (3) *EfficientNet-B0* Tan & Le (2019), and (4) *ViT-B/16* Dosovitskiy et al. (2020). These models span both convolutional and transformer-based paradigms and offer a clear comparison across varying robustness profiles and architectural biases. For all models, we report clean and attacked accuracies under each defense method, using the same attack configuration and patch size distribution.

**Datasets and Attacks.** We evaluate on both synthetic and real-world patch benchmarks: (1) *ImageNet-Patch* Pintor (2023), a 50/50 balanced dataset of attacked and clean ImageNet samples, comprising 400 test images, where attacks are applied to exactly 50% of the data to ensure balanced evaluation; and (2) *APRICOT* Liu et al. (2022), a real-world dataset of 873 images, each containing a physically applied adversarial patch.

We test two strong attacks: the classical adversarial patch Brown et al. (2017) targeting CNNs, and PatchFool Fu (2022) targeting vision transformers. Patches are randomly placed and vary in size from $25 \times 25$ to $65 \times 65$.

**Defense Mechanisms.** We compare against several methods: (1) JPEG compression Dziugaite et al. (2016), (2) Spatial smoothing Xu et al. (2017), (3) SAC Liu et al. (2022), and (4) DIF-Fender Kang et al. (2024), a recent diffusion-based approach. We also evaluate a retrieval-only baseline that flags regions as adversarial based on visual similarity, without using VLM reasoning.

**Evaluation Protocol.** On *ImageNet-Patch*, we report classification accuracy over a balanced 50/50 clean/attacked split. On *APRICOT*, we report binary accuracy (presence vs. absence of a patch) across three settings: (1) *Clean*, (2) *Undefended*, and (3) *Defended*. Candidate regions are retrieved using top-$k = 2$ cosine similarity and verified via VLM prompts. Thresholds are calibrated on a held-out validation set to ensure fair comparisons across all methods.

## 6 RESULTS

Table 1 presents the accuracy performance of various defense mechanisms on the APRICOT dataset Braunegg et al. (2020) under adversarial patch attacks of varying sizes ($25 \times 25$ to $65 \times 65$). Traditional defenses such as JPEG compression Dziugaite et al. (2016), spatial smoothing Xu et al. (2017), and SAC Liu et al. (2022) provide only modest improvements, particularly as patch size increases. DIFFender Kang et al. (2024) shows stronger robustness, achieving better accuracy across all patch sizes.

Table 1: Accuracy (%) on APRICOT Braunegg et al. (2020) with adversarial patches of varying sizes. Methods are evaluated in 0-shot (0S), 2-shot (2S), and 4-shot (4S) settings; methods without few-shot use show "–". Values prefixed by * indicate the best 0S result, underlined rows the second-best method, and **bold rows** the best overall methods.

| Method | 25 × 25 | | | 50 × 50 | | | 55 × 55 | | | 65 × 65 | | |
|---|---|---|---|---|---|---|---|---|---|---|---|---|
| | 0S | 2S | 4S | 0S | 2S | 4S | 0S | 2S | 4S | 0S | 2S | 4S |
| Undefended | 34.59 | – | – | 32.18 | – | – | 30.24 | – | – | 28.55 | – | – |
| JPEG Dziugaite et al. (2016) | 29.35 | – | – | 32.53 | – | – | 35.28 | – | – | 41.11 | – | – |
| Spatial Smoothing Xu et al. (2017) | 33.56 | – | – | 36.19 | – | – | 39.17 | – | – | 42.26 | – | – |
| SAC Liu et al. (2022) | 45.93 | – | – | 48.22 | – | – | 49.14 | – | – | 52.80 | – | – |
| DIFFender Kang et al. (2024) | *65.06 | – | – | *66.32 | – | – | *68.61 | – | – | *70.90 | – | – |
| Baseline | 56.81 | – | – | 59.56 | – | – | 60.59 | – | – | 69.64 | – | – |
| **Ours (Qwen-VL-Plus)** | 45.37 | 76.18 | 87.64 | 46.40 | 77.90 | 88.78 | 47.55 | 79.62 | 90.50 | 50.98 | 81.91 | 92.22 |
| **Ours (Qwen2.5-VL-72B)** | 47.37 | 78.18 | 89.64 | 48.40 | 79.90 | 90.78 | 49.55 | 81.62 | 92.50 | 52.98 | 83.91 | 94.22 |
| **Ours (UI-TARS-72B-DPO)** | 49.37 | 80.18 | 91.64 | 50.40 | 81.90 | 92.78 | 51.55 | 83.62 | 94.50 | 54.98 | 85.91 | 96.22 |
| **Ours (Gemini)** | 56.24 | 82.59 | **93.92** | 57.16 | 85.11 | **96.33** | 58.76 | 86.94 | **96.79** | 63.12 | 90.26 | **97.93** |

Our approach not only consistently outperforms these 0-shot baselines but also shows increasing advantage as patch size grows-demonstrating better scalability under high-strength attacks. Notably, even the 0-shot version of our method achieves competitive results, while the 4-shot configuration delivers substantial gains, outperforming all baselines by a large margin. It maintains high classification accuracy even under challenging conditions, as shown in the confusion matrix visualizations in Appendix 9, reinforcing the effectiveness of combining retrieval with generative vision-language reasoning in real-world adversarial settings.

Table 2: Accuracy (%) of four models under adversarial patch attacks of varying sizes. Each method is evaluated under three configurations: 0-shot (0S), 2-shot (2S), and 4-shot (4S), reflecting increasing levels of visual context provided to the vision-language model. For methods that do not support few-shot adaptation, results for 2S and 4S are omitted and marked with "–". Values prefixed by * indicate the best-performing method in the 0-shot setting, underline highlights the second-best overall result across all configurations, and **bold** denotes the highest overall accuracy. This presentation enables a clear comparison of zero- and few-shot performance across varying patch sizes and models.

| Model | Method | Clean | 25 × 25 | | | 50 × 50 | | | 55 × 55 | | | 65 × 65 | | |
|---|---|---|---|---|---|---|---|---|---|---|---|---|---|---|
| | | | 0S | 2S | 4S | 0S | 2S | 4S | 0S | 2S | 4S | 0S | 2S | 4S |
| ResNet-50 He et al. (2016) | Undefended | | 7.50 | – | – | 9.25 | – | – | 8.75 | – | – | 6.95 | – | – |
| | JPEG Dziugaite et al. (2016) | | 50.75 | – | – | 51.75 | – | – | 49.25 | – | – | 49.00 | – | – |
| | Spatial Smoothing Xu et al. (2017) | | 55.50 | – | – | 58.25 | – | – | 55.25 | – | – | 50.75 | – | – |
| | SAC Liu et al. (2022) | | *64.75 | – | – | *66.75 | – | – | *68.00 | – | – | *69.50 | – | – |
| | Baseline | 97.50 | 58.50 | – | – | 59.75 | – | – | 62.00 | – | – | 62.50 | – | – |
| | **Ours (Qwen-VL-Plus)** | | 49.75 | 70.00 | 85.25 | 54.00 | 73.00 | 86.50 | 62.50 | 75.00 | 87.25 | 60.00 | 79.50 | 88.00 |
| | **Ours (Qwen2.5-VL-72B)** | | 55.25 | 82.00 | 88.25 | 58.25 | 84.00 | 60.25 | 59.50 | 86.00 | 90.50 | 60.00 | 91.25 | 91.50 |
| | **Ours (UI-TARS-72B-DPO)** | | 54.50 | 83.00 | 89.50 | 55.50 | 87.75 | 90.50 | 57.50 | 86.25 | 89.75 | 57.50 | 87.50 | 94.00 |
| | **Ours (Gemini)** | | 56.25 | 87.25 | **93.25** | 58.50 | 89.75 | **93.75** | 59.75 | 90.25 | **96.25** | 60.25 | 91.25 | **99.25** |
| ResNeXt-50 Xie et al. (2017) | Undefended | | 9.25 | – | – | 11.00 | – | – | 10.75 | – | – | 8.95 | – | – |
| | JPEG Dziugaite et al. (2016) | | 48.75 | – | – | 50.75 | – | – | 47.75 | – | – | 46.50 | – | – |
| | Spatial Smoothing Xu et al. (2017) | | 55.75 | – | – | 57.50 | – | – | 55.75 | – | – | 50.25 | – | – |
| | SAC Liu et al. (2022) | | *64.75 | – | – | *66.25 | – | – | *68.00 | – | – | *66.75 | – | – |
| | Baseline | 97.50 | 56.50 | – | – | 58.50 | – | – | 60.25 | – | – | 61.75 | – | – |
| | **Ours (Qwen-VL-Plus)** | | 48.25 | 68.50 | 83.00 | 52.00 | 71.25 | 84.50 | 58.00 | 72.75 | 85.25 | 60.25 | 77.00 | 86.25 |
| | **Ours (Qwen2.5-VL-72B)** | | 53.25 | 78.25 | 85.75 | 58.50 | 80.75 | 87.00 | 59.50 | 84.00 | 88.25 | 60.25 | 90.00 | 90.75 |
| | **Ours (UI-TARS-72B-DPO)** | | 52.50 | 80.75 | 85.75 | 55.25 | 85.00 | 89.25 | 55.25 | 86.25 | 91.00 | 59.25 | 84.75 | 93.25 |
| | **Ours (Gemini)** | | 55.50 | 85.00 | **91.25** | 57.75 | 87.50 | **92.75** | 58.75 | 88.50 | **94.75** | 60.75 | 89.75 | **98.50** |
| EfficientNet Tan & Le (2019) | Undefended | | 24.25 | – | – | 25.75 | – | – | 24.00 | – | – | 21.50 | – | – |
| | JPEG Dziugaite et al. (2016) | | 51.00 | – | – | 53.75 | – | – | 50.75 | – | – | 49.25 | – | – |
| | Spatial Smoothing Xu et al. (2017) | | *60.50 | – | – | *63.25 | – | – | 61.75 | – | – | 57.50 | – | – |
| | SAC Liu et al. (2022) | | 58.25 | – | – | 60.75 | – | – | *63.25 | – | – | *67.25 | – | – |
| | Baseline | 95.50 | 54.75 | – | – | 56.75 | – | – | 58.25 | – | – | 61.00 | – | – |
| | **Ours (Qwen-VL-Plus)** | | 50.25 | 69.25 | 84.00 | 53.00 | 72.25 | 85.50 | 58.00 | 74.00 | 86.25 | 58.00 | 78.75 | 87.75 |
| | **Ours (Qwen2.5-VL-72B)** | | 54.50 | 79.50 | 87.00 | 59.25 | 82.00 | 89.00 | 57.00 | 85.00 | 59.50 | 60.50 | 91.00 | 92.00 |
| | **Ours (UI-TARS-72B-DPO)** | | 49.75 | 80.50 | 85.25 | 52.25 | 83.00 | 88.75 | 54.75 | 82.75 | 91.00 | 57.75 | 86.00 | 95.00 |
| | **Ours (Gemini)** | | 53.00 | 84.25 | **91.25** | 55.50 | 85.75 | **93.50** | 57.00 | 88.00 | **95.75** | 59.75 | 89.75 | **97.50** |
| ViT-B-16 Fu (2022) | Undefended | | 27.75 | – | – | 29.25 | – | – | 27.00 | – | – | 24.25 | – | – |
| | JPEG Dziugaite et al. (2016) | | 57.75 | – | – | 58.75 | – | – | 55.50 | – | – | 51.00 | – | – |
| | Spatial Smoothing Xu et al. (2017) | | *66.75 | – | – | *67.25 | – | – | 64.00 | – | – | 61.25 | – | – |
| | SAC Liu et al. (2022) | | 63.25 | – | – | 64.75 | – | – | *65.75 | – | – | *69.25 | – | – |
| | Baseline | 97.75 | 59.50 | – | – | 61.50 | – | – | 62.75 | – | – | 64.00 | – | – |
| | **Ours (Qwen-VL-Plus)** | | 51.25 | 69.50 | 84.25 | 55.00 | 72.50 | 85.75 | 60.50 | 76.00 | 86.75 | 63.00 | 79.00 | 87.25 |
| | **Ours (Qwen2.5-VL-72B)** | | 56.75 | 78.75 | 87.00 | 60.75 | 81.00 | 88.75 | 61.00 | 84.50 | 90.50 | 60.25 | 91.00 | 91.75 |
| | **Ours (UI-TARS-72B-DPO)** | | 53.25 | 82.00 | 89.50 | 54.75 | 84.25 | 91.00 | 56.75 | 85.50 | 93.25 | 59.50 | 88.75 | 95.25 |
| | **Ours (Gemini)** | | 58.75 | 86.75 | **93.50** | 60.75 | 89.00 | **95.25** | 61.25 | 90.75 | **98.75** | 63.00 | 93.00 | **99.00** |

Table 2 reports defense accuracy under adversarial patch attacks of varying sizes. As expected, performance drops sharply without defense. Traditional methods like JPEG compression Dziugaite et al. (2016), spatial smoothing Xu et al. (2018), and SAC Liu et al. (2022) show limited robustness, while DIFFender Kang et al. (2024) performs better through generative reconstruction.

Our retrieval-only baseline outperforms these, highlighting the value of visual similarity. The full method-combining retrieval with VLM reasoning-achieves the best results, with the 4-shot variant nearly restoring clean accuracy under large patches. This demonstrates the effectiveness of retrieval-augmented generative reasoning for adaptive patch detection.

We further analyze the effect of prompt design on detection performance. As shown in Appendix 8, incorporating visual examples of both adversarial patches and attacked images into the prompt improves detection accuracy, with the combined prompt format achieving the best results across multiple models and patch sizes. This finding highlights the importance of structured, context-rich prompts in maximizing the reasoning capabilities of VLMs. Specifically, prompts that present both the cause (adversarial patch) and the effect (altered image behavior) enable the VLM to better associate visual cues with adversarial intent, even in zero-shot settings. This insight suggests that prompt engineering is not merely a cosmetic component but a critical design factor in VLM-driven adversarial detection pipelines. It also opens the door to automated or learned prompt optimization strategies that could further boost performance under different deployment scenarios. Additionally, Appendix A.8 presents an ablation study that quantifies the impact of key system components. We analyze the trade-offs introduced by retrieval strategy choices (e.g., key/value formulation, embedding granularity), prompt formulations (e.g., descriptive vs. direct), few-shot context sizes (0-shot, 2-shot, 4-shot), and inference-time efficiency. These experiments offer actionable insights into which design choices yield the best accuracy-performance trade-off and help identify bottlenecks in system scalability. Together, these findings reinforce the critical role of retrieval and prompt design in enabling robust, generalizable adversarial patch detection without the need for retraining.

## 7 DISCUSSION AND CONCLUSION

We introduced a training-free framework for adversarial patch detection that integrates visual retrieval-augmented generation with vision-language models. By leveraging a precomputed and expandable database of diverse adversarial patches, our method enables dynamic retrieval and context-aware reasoning without any model retraining or fine-tuning. This makes our approach both scalable and deployment-ready in dynamic or resource-constrained environments. In contrast to many prior defenses that rely on task-specific training regimes or assumptions about patch geometry, our method generalizes effectively to a broad range of patch types-including naturalistic, camouflaged, and physically realizable attacks.

Extensive evaluations on two complementary datasets-*ImageNet-Patch*, a synthetic benchmark with clean/attacked image pairs, and *APRICOT*, a real-world dataset with 873 physically attacked images-demonstrate the robustness of our framework. Across varying patch sizes and attack methods, our method consistently outperforms traditional defenses such as JPEG compression Dziugaite et al. (2016), spatial smoothing Xu et al. (2017), SAC Liu et al. (2022), and DIFFender Kang et al. (2024). Our full system achieves detection rates of up to 98%, and crucially, maintains performance as the threat severity increases.

We also report inference-time performance and parallelization trade-offs to assess real-world feasibility. Appendix 8 and Appendix A.6 provide qualitative comparisons, confusion matrices, and generalization analysis to diverse patch shapes, further reinforcing the robustness of our method.

**Limitations and Future Work.** While effective, our method currently assumes access to a representative patch database. Future work will focus on automatically identifying and augmenting missed or novel adversarial patterns using generative models and self-supervised learning. We also aim to incorporate uncertainty quantification into VLM outputs to better handle ambiguous or borderline cases. Furthermore, improving inference speed-particularly for high-resolution images and real-time applications-remains an important direction for deployment at scale.

**Conclusion.** Our VRAG-based framework combines retrieval-based search with generative vision-language reasoning to offer a robust, adaptive, and training-free solution to adversarial patch detection. It achieves high accuracy, generalizes across patch types, and requires minimal supervision-making it a practical and scalable defense strategy for modern vision systems. Furthermore, our method successfully generalized across diverse patches within the database and reliably detected novel attack scenarios.

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

# A APPENDIX: ABLATION STUDY

We perform all evaluations on the ImageNet-Patch Pintor (2023) dataset.

## A.1 EFFECT OF PARALLELIZATION

Parallelization significantly improves the efficiency of adversarial patch database creation. Since the application of each patch to each image-and the subsequent embedding computation-are independent operations, the process can be parallelized across multiple workers Kazoom et al. (2022). This enables rapid generation and encoding of large-scale patched image datasets.

In our setup, we applied adversarial patches to a collection of clean images, using a key-value approach where each image was divided into a $5 \times 5$ grid. Patch embeddings served as keys, while embeddings of image regions acted as values for retrieval. The end result was a database of 3,500 patch-image pairs with corresponding embeddings. To evaluate scalability, we measured execution time with varying levels of parallelism, confirming substantial speedups as the number of workers increased.

Table 3: Execution time for adversarial patch detection with different numbers of workers. Results are reported as mean $\pm$ standard deviation, in minutes.

| Number of Workers | Execution Time (min) |
| --- | --- |
| 1 | $24.57 \pm 0.07$ |
| 2 | $12.12 \pm 0.10$ |
| 3 | $8.11 \pm 0.16$ |
| 4 | $6.14 \pm 0.26$ |
| 5 | $4.59 \pm 0.40$ |
| 6 | $3.58 \pm 0.54$ |

As shown in 3, using a single worker resulted in an average execution time of 24.57 minutes, whereas increasing the number of workers to six reduced the execution time to 3.58 minutes, demonstrating a 6.86× speedup. The results indicate that distributing the workload across multiple processes significantly reduces execution time while maintaining detection accuracy.

These findings validate the effectiveness of parallelization in our method, allowing it to scale efficiently for larger datasets. The speedup enables the rapid processing of extensive adversarial patch collections, making real-time detection feasible.

## A.2    EMBEDDING DISTANCE ANALYSIS

We evaluate the effectiveness of our retrieval mechanism through an ablation study comparing several distance metrics for nearest-neighbor retrieval, including cosine similarity, L1 distance, L2 distance, and Wasserstein distance. All experiments in this subsection were conducted on the ImageNet-Patch dataset. Rather than relying solely on cosine similarity for retrieving stored adversarial patches, we also assess alternative metrics using embeddings extracted via CLIP Radford et al. (2021).

Given an input image $I$, we partition it into grid-based regions and extract feature embeddings using CLIP's image encoder:

$$E_I = f(I), \quad E_{\mathcal{D}} = \{f(D_i) \mid D_i \in \mathcal{D}\}, \tag{9}$$

where $f(\cdot)$ denotes the CLIP embedding function and $\mathcal{D}$ represents the precomputed adversarial patch database.

For cosine similarity-based retrieval, the similarity score is computed as:

$$S(E_I, E_{\mathcal{D}}) = \frac{E_I \cdot E_{\mathcal{D}}}{\|E_I\| \, \|E_{\mathcal{D}}\|}, \tag{10}$$

with a stored adversarial patch retrieved if $S(E_I, E_{\mathcal{D}})$ exceeds a similarity threshold $\tau_s$.

We also evaluate L1 and L2 distances. The L1 distance is defined as:

$$d_{\text{L1}}(E_I, E_{\mathcal{D}}) = \sum |E_I - E_{\mathcal{D}}|, \tag{11}$$

and the L2 distance is given by:

$$d_{\text{L2}}(E_I, E_{\mathcal{D}}) = \|E_I - E_{\mathcal{D}}\|_2. \tag{12}$$

For both L1 and L2 distances, retrieval is triggered when the computed distance falls below a threshold ($\tau_{\text{L1}}$ or $\tau_{\text{L2}}$, respectively).

Additionally, we examine the Wasserstein distance, which measures the optimal transport cost between distributions. For two distributions $P$ and $Q$ over the embedding space, the Wasserstein distance is defined as:

$$W(E_I, E_{\mathcal{D}}) = \inf_{\gamma} \mathbb{E}_{(x,y)\sim\gamma}[\|x - y\|], \tag{13}$$

where $\gamma$ is a joint distribution with marginals $P$ and $Q$. This metric quantifies the minimal effort required to transport mass between the two embedding distributions.

We compare the retrieval effectiveness of these four metrics using Gemini-2.0 DeepMind (2024) for final classification. The cosine similarity-based approach achieves the highest classification accuracy at 98.00%, followed by L2 distance (89.75%), L1 distance (86.25%), and Wasserstein distance (84.25%). These results are visualized in 3.

These results indicate that cosine similarity most effectively captures the high-dimensional semantic relationships essential for robust adversarial patch retrieval, while the alternative metrics, although reasonable, perform less effectively-particularly the Wasserstein distance, which struggles to model distributional similarity from limited embedding samples.

## A.3    INFERENCE TIME ANALYSIS

All experiments in this subsection were conducted on the ImageNet-Patch dataset. In addition to detection performance, we assess the inference time required for each defense mechanism. For an

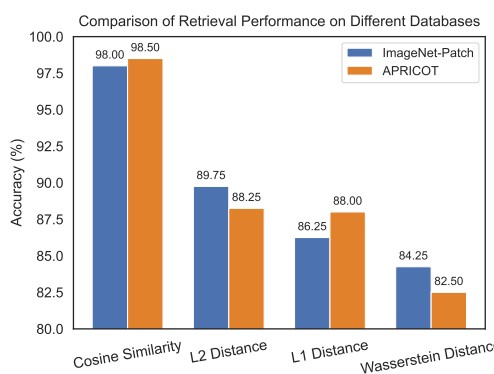 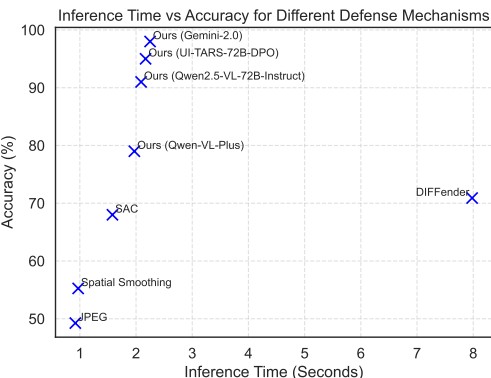

Figure 3: Retrieval performance using different distance metrics on CLIP embeddings.

Figure 4: Inference time vs. accuracy for different defense mechanisms.

input image $I$, the processing time for a defense mechanism $D$ is defined as:

$$T_D = \frac{1}{N} \sum_{i=1}^{N} t_i, \tag{14}$$

where $t_i$ is the processing time for the $i$-th image and $N$ is the total number of test images.

We analyze the trade-off between inference time $T_D$ and classification accuracy $A_D$, which is calculated as:

$$A_D = \frac{C_{\text{correct}}}{C_{\text{total}}} \times 100, \tag{15}$$

with $C_{\text{correct}}$ representing the number of correctly classified images and $C_{\text{total}}$ the total number of images.

As shown in 4, JPEG compression Dziugaite et al. (2016) and Spatial Smoothing Xu et al. (2017) offer the fastest inference times (0.92s and 0.97s, respectively), albeit with limited accuracy improvements (49.25% and 55.25%). SAC Liu et al. (2022) requires 1.58s while achieving an accuracy of 68.00%, and DIFFender Kang et al. (2024) attains an accuracy of 70.90% with an inference time of 7.98s.

Our method, leveraging Qwen-VL-Plus Cloud (2023), Qwen2.5-VL-72B-Instruct Cloud (2024), UI-TARS-72B-DPO Research (2024), and Gemini-2.0 DeepMind (2024), achieves superior classification accuracy (79.00%, 91.00%, 95.00%, and 98.00%, respectively) with inference times of 1.97s, 2.09s, 2.17s, and 2.25s.

These findings highlight a clear performance–efficiency trade-off: higher detection accuracy generally demands increased computational cost. Our approach effectively balances these aspects by leveraging retrieval-augmented detection while maintaining inference times that remain competitive with existing defense mechanisms.

## A.4 PROMPT ENGINEERING ANALYSIS

All experiments in this subsection were conducted on the ImageNet-Patch dataset. To investigate the impact of prompt design Gu et al. (2023) on adversarial patch detection, we conducted an ablation study evaluating five distinct prompting strategies. Each strategy aims to guide the VLM in classifying whether an image contains an adversarial patch. Given an input image $I$, the VLM is provided with a textual prompt $\mathcal{T}$ and returns a classification response:

$$R = \mathcal{V}(\mathcal{T}, I), \tag{16}$$

where $\mathcal{V}$ represents the VLM inference function.

To enhance context, we leverage a retrieved set of adversarial patch examples $\{P_1, \ldots, P_m\}$, where each $P_i$ is an adversarial patch stored in the database, and a set of attacked images $\{I_1, \ldots, I_k\}$, where each $I_j$ is a full image containing an applied adversarial patch. These elements provide additional visual references during inference.

The prompting strategies evaluated are as follows, along with the specific examples used:

1. **Instruction-only:** A generic instruction without examples:

   ```
   ''Adversarial physical attacks involve placing random
   patches on images.  You are an expert in identifying
   such patches.  Is the following image attacked?
   Answer 'yes' or 'no'.''
   ```

2. **Attacked Images:** The instruction followed by examples of attacked images $\{I_1, \ldots, I_k\}$:

   ```
   ''Here are examples of images that have been
   attacked:  [Image 1], [Image 2], [Image 3].  Given
   the next image, is it attacked?  Answer 'yes' or
   'no'.''
   ```

3. **Patch Examples:** The instruction followed by examples of extracted adversarial patches $\{P_1, \ldots, P_m\}$:

   ```
   ''Here are examples of adversarial patches:  [Patch
   1], [Patch 2], [Patch 3].  Given the next image, is
   it attacked?  Answer 'yes' or 'no'.''
   ```

4. **Chain-of-Thought (CoT):** The instruction augmented with reasoning:

   ```
   ''Adversarial attacks often involve adding suspicious
   patches.  First, analyze if there are irregular
   regions.  Then, decide if an attack is present.
   Is the following image attacked?  Answer 'yes' or
   'no'.''
   ```

5. **Combined (Final, Without CoT):** The instruction with both attacked images and patch examples:

   ```
   ''Adversarial physical attacks involve random patches
   on images.  You are an expert at detecting them.
   Here are examples of adversarial patches:  [Patch
   1], [Patch 2].  Here are examples of attacked images:
   [Image 1], [Image 2].  Given the above context, is
   this image attacked?  Please answer 'yes' or 'no'.''
   ```

To quantify the effectiveness of each prompt type, we measured the detection accuracy $A_{\mathcal{T}}$ obtained under each configuration. The final selected prompt, as presented in 2, corresponds to the Combined (Final) strategy, which achieved the highest detection accuracy of 98.00%. The complete results are summarized in 5, where we observe that simple instructional prompts result in low accuracy (58.00%), while adding contextual examples (patches and attacked images) significantly improves performance. The CoT-based prompt further enhances accuracy to 91.25%, whereas the combined strategy achieves the highest overall detection rate.

This ablation study highlights that careful prompt engineering, particularly including few-shot visual examples and reasoning, is critical for maximizing VLM-based adversarial patch detection.

### A.5 IMPACT OF FEW-SHOT CONTEXT SIZE ON CLASSIFICATION ACCURACY

All experiments in this subsection were conducted on the ImageNet-Patch dataset. To evaluate the effect of context size on adversarial patch detection, we conducted an ablation study by varying the number of few-shot examples provided to the VLM during inference. Let $k \in \{0, 1, \ldots, 6\}$ denote the number of retrieved examples (i.e., the few-shot shots). For each $k$-shot configuration, we measured the classification accuracy $A_k$ of the VLM in detecting adversarial patches across four different models: Qwen-VL-Plus, Qwen2.5-VL-Instruct, UI-TARS-72B-DPO, and Gemini-2.0.

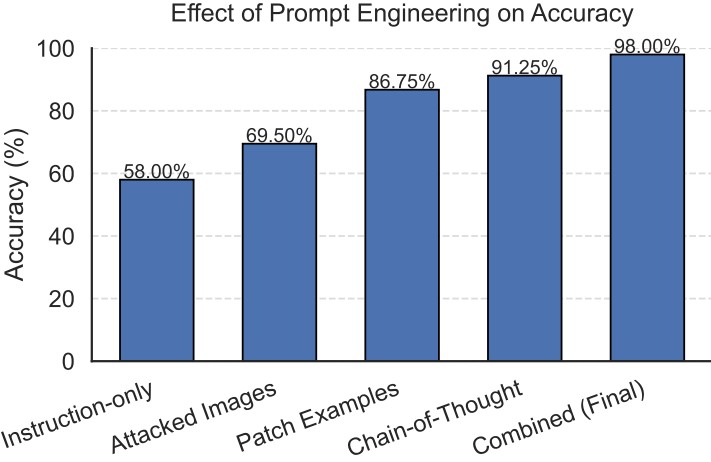

Figure 5: Effect of prompt engineering on adversarial patch classification accuracy.

6 illustrates the trend of $A_k$ as a function of $k$. Across all models, we observe a consistent improvement in detection accuracy with increasing values of $k$, indicating that providing more contextual examples strengthens the model's ability to generalize and distinguish adversarial patterns. Notably, UI-TARS-72B-DPO consistently achieves intermediate performance, surpassing Qwen-based models and closely approaching Gemini-2.0 accuracy.

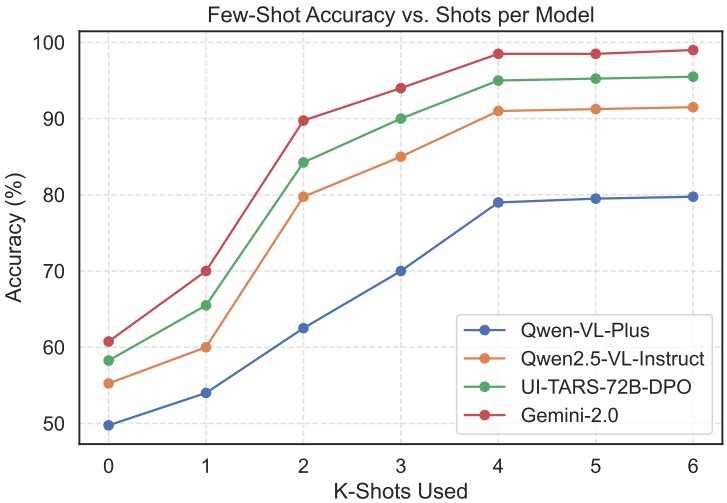

Figure 6: Few-shot detection accuracy across varying context sizes $k$.

These results suggest that larger few-shot contexts allow the VLM to better align the input query with prior adversarial patterns stored in the retrieval database. However, the performance gains tend to plateau beyond $k = 4$, highlighting a saturation effect where additional examples yield diminishing returns. The comparison also reveals that more capable VLMs (e.g., Gemini-2.0 and UI-TARS-72B-DPO) benefit more rapidly from few-shot conditioning than smaller models such as Qwen-VL-Plus and Qwen2.5-VL-Instruct, although Gemini-2.0 still demonstrates superior performance overall.

## A.6 GENERALIZATION TO DIVERSE PATCH SHAPES

Real-world adversarial patches appear in many shapes and textures, from geometric (square, round, triangular) to naturalistic or camouflage-like forms. To ensure robustness against these diverse pat-

terns, we incorporate a range of patch types in the database creation phase. Concretely, each patch $P_i \in \mathcal{P}$ may be:

$$\text{square, round, triangle, realistic,} \ldots$$

Since detection relies on embedding-based similarity rather than geometric assumptions, unusual or irregular patch shapes remain identifiable as long as their embeddings lie above a retrieval threshold $\tau$. In practice, this approach allows our VRAG-based framework to detect both canonical patches and highly unobtrusive, adaptive adversarial artifacts designed to evade simpler defenses.

By collectively leveraging a rich database of patch embeddings, a retrieval-augmented paradigm, and a capable vision-language model, our method achieves robust generalization in adversarial patch detection across a wide spectrum of attack strategies.

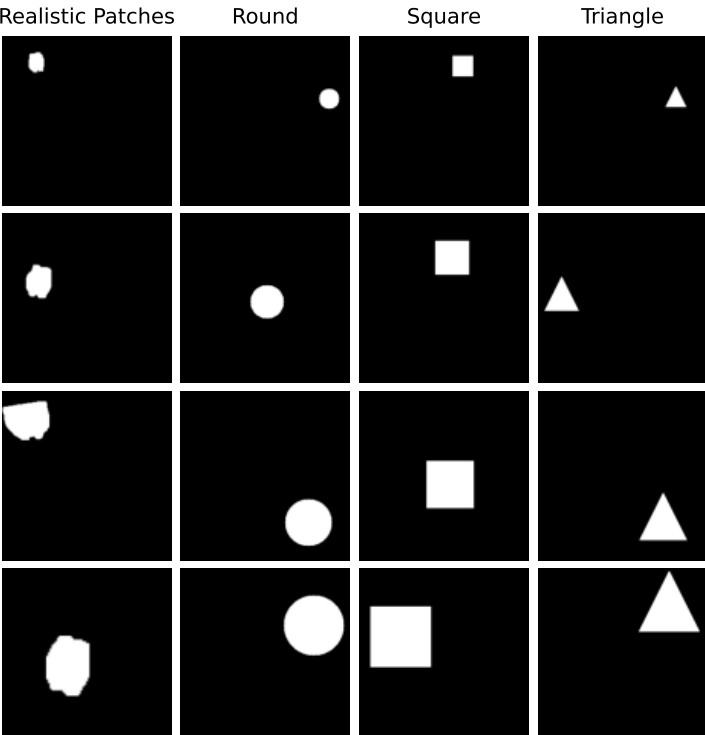

Figure 7: Examples of adversarial patch masks used in our dataset. We consider four types: realistic, round, square, and triangle. This diversity improves robustness across patch shapes.

## A.7 QUALITATIVE RESULTS

In addition to quantitative evaluations, we present qualitative results highlighting the effectiveness of our proposed framework compared to existing defenses. 8 illustrates visual comparisons across various defense mechanisms: Undefended, JPEG compression Dziugaite et al. (2016), Spatial Smoothing Xu et al. (2017), SAC Liu et al. (2022), DIFFender Kang et al. (2024), and our method.

Adversarial patches remain clearly visible and disruptive in both Undefended and JPEG-compressed images, indicating that these methods fail to mitigate patch attacks effectively. SAC partially reduces the visibility of adversarial patches but does not consistently eliminate them, often leaving residual disruptions. DIFFender Kang et al. (2024) demonstrates improved effectiveness compared to SAC by further reducing patch visibility, though residual disturbances remain apparent.

In contrast, our method reliably identifies and neutralizes adversarial patches, effectively mitigating their influence while preserving image integrity. However, our approach also has specific failure modes, particularly evident when the adversarial patch blends seamlessly into the noisy background of an image, matching its distribution. In such challenging cases (e.g., the last row of the right-hand

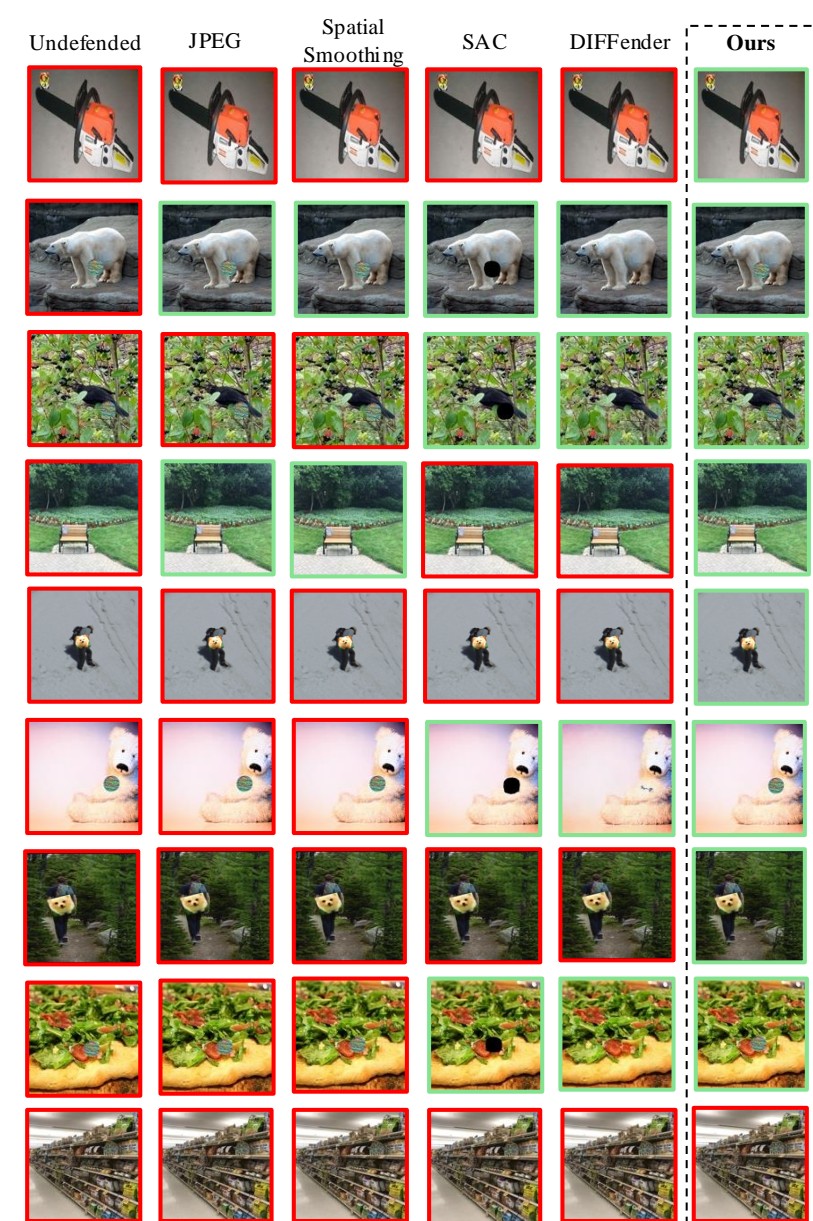

Figure 8: Qualitative comparison of different defense mechanisms. From left to right: Undefended, JPEG compression Dziugaite et al. (2016), Spatial Smoothing Xu et al. (2017), SAC Liu et al. (2022), DIFFender Kang et al. (2024) and our method.

table in 8), the model may struggle to accurately differentiate between patch and background noise, highlighting a limitation to be addressed in future research.

## A.8 IMPACT OF FEW-SHOT RETRIEVAL ON VLM ACCURACY

To further understand performance across different vision-language models (VLMs), 9 shows confusion matrices for Qwen-VL-Plus Bai et al. (2024), Qwen2.5-VL-72B Bai et al. (2025), UI-TARS-72B-DPO Research (2024), and Gemini-2.0 Team et al. (2023) under 0-shot, 2-shot, and 4-shot configurations. Increasing the number of retrieved examples consistently improves both true-positive

and true-negative rates. Notably, the 4-shot configuration with Gemini-2.0 yields near-perfect separation between adversarial and clean samples. While Gemini-2.0 remains the top-performing model, UI-TARS-72B-DPO achieves highly competitive results, outperforming all other open-source VLMs by a significant margin.

These findings highlight the power of retrieval-augmented prompting for adversarial patch detection-especially when representative visual-textual context is injected via advanced VLMs.

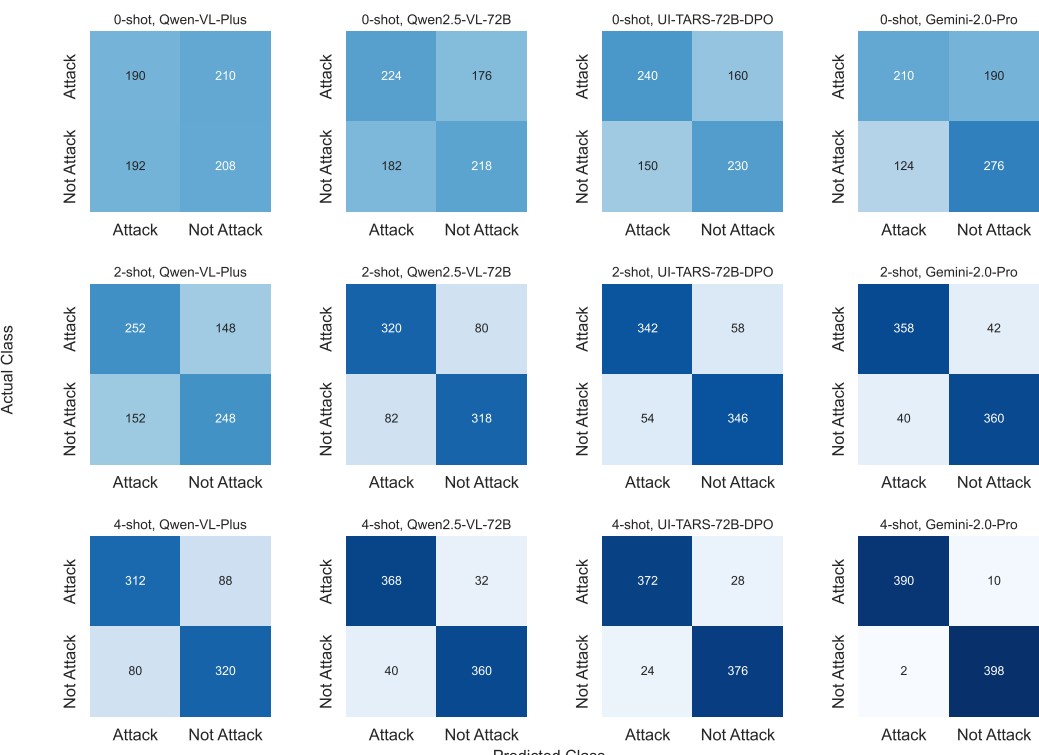

Figure 9: Confusion matrices across three few-shot configurations (rows) and four VLMs (columns). Axes represent predicted and actual classes ("Attack" vs. "Not Attack"). Gemini-2.0 achieves the best overall accuracy, while UI-TARS-72B-DPO offers the strongest open-source performance.

## LLM USAGE DISCLOSURE

In accordance with the ICLR 2026 policy on LLM usage, we disclose that a large language model was utilized during the preparation of this manuscript. The use of the LLM was strictly limited to correcting grammar and checking code syntax. All research contributions, experimental design, analysis, and the core text were generated by the human authors. The authors have reviewed all AI-assisted content and bear full responsibility for the final submission.

