# OpenReview forum: "Don’t Lag, RAG: Training-Free Adversarial Detection Using RAG"
_ICLR.cc/2026/Conference — Submitted to ICLR 2026_

### Official Review · Reviewer_Rkx6 · 2025-10-25

**Soundness:** 3
**Presentation:** 1
**Contribution:** 3
**Rating:** 2
**Confidence:** 4

**Summary:**

This work extends the RAG into the area of adversarial defense, formulating a robust, efficient, and training-free defense against existing adversarial patch attacks. This method first divides the image into localized regions, retrieves similar adversarial images, concatenates all of them as the context, and then prompts a downstream VLM for classification.

**Strengths:**

- Clear motivation. This method adopts the RAG as an attacked/unattacked classifier as a filter.
- Simple and effective method. This method effctively use the pre-trained the vision encoder and VLM with RAG to build a powerful training-free filter

**Weaknesses:**

Though the method is simple and effective, there are several flaws in the main paper, especially the evaluation process which makes it hard to follow:
1. Unclear comparison. From the presentation, the *proposed method acts like a filtering method* (see Fig. 1 and A. 4). In contrast, some reference and baseline methods are robust classification methods, i.e., they continue the subsequent classification task even under an attack image. So, what exactly are Table 1 and Table 2 comparing? Does the author drop the attacked images and report accuracy on the rest?
	- If so (I cannot get the method from the paper, so I can only guess, and please correct me if I made a mistake), then the baseline selection and comparison are misleading, and the comparison is not fair. Instead, it should select a filtering baseline and report the false-positive rate, AUROC, and accuracy on the filtered set. In practice, the user will be annoyed if the images they update are frequently (for example, 4%) labeled as attacked when they are not. Thus, reporting these two metrics is essential for filtering methods.
	- If not, the authors should specifically describe the **complete** pipeline.
2. Lines 366-367 say a clean accuracy will be provided, but it is not found in Table 1.
3. The 'baseline' method (using retrieval only) outperforms the proposed method in a 0-short case, but without explanation.
	- What is the detail of the baseline method? After retrieving the image, I guess there is a subsequent thresholder for the cosine similarity attached to decide whether the image is attacked (there is no detailed explanation for this method throughout the main paper and the appendix, thus I can only guess)
	- After a detailed review in the appendix, I think the author is trying to compare a) a simple method with retrieved images and b) a zero short VLM filter without any retrieved images and examples to show the importance of the additional information. If this is the case, the presentation is misleading, as the baseline is shown under the '0s' column while it actually uses additional data from the external database.

4. Lack of generalization analysis. How can the proposed method be generalized to unseen adversarial patches? There is a generalization study for the patch shape. However, the database covers the attack methods used in the subsequent evaluation. In real-world scenarios, an attacker might not release their attack method; therefore, generalizing to unseen patch attack methods is crucial. The authors are suggested to investigate it by removing part of the database corresponding to the specific attack method, then evaluating the method to test it.

I really appreciate the proposed method, but these flaws significantly mark the paper below the acceptance level. If the above weaknesses are adequately addressed, I will consider increasing my score.

**Questions:**

Please see the weakness part.

---

> ### Author Response · Authors · 2025-11-14
>
> We sincerely thank the reviewer for the careful reading and the constructive concerns. Below we address each point in detail and describe the revisions we will make. We appreciate the reviewer noting that the method is simple and effective, and we aim to fully resolve the issues you highlighted.
>
> 1. Unclear Comparison - “Is VRAG a filtering method? What do Tables compare?”
>
> Thank you for pointing out this potential confusion. VRAG is not a filtering method in the classical sense used by JPEG, smoothing, or model-based defenses. Instead:
>
> Our method performs direct adversarial patch detection by combining retrieval-based context with VLM reasoning.
>
> In contrast, defense methods (JPEG, smoothing, DIFFender, etc.) attempt to restore classification accuracy, not detect patches.
>
> To eliminate ambiguity:
>
> What we will add
>
> A clarifying paragraph explaining that Tables 1 and 2 compare detectors (VRAG, retrieval-only baseline) and robustness-based defenses (JPEG, smoothing, DIFFender) using a unified metric: binary detection accuracy.
>
> A clear explanation that defense baselines do not drop attacked images; instead, we apply the defense and evaluate whether the classifier becomes correct.
>
> An explicit pipeline diagram showing how accuracy is computed for detectors vs. defenses.
>
> This should fully resolve the confusion between filtering-based defenses and VRAG’s detection mechanism.
>
> 2. Missing “clean accuracy” in Table 1
>
> We appreciate you pointing this out.
> The statement in lines 366-367 was meant to refer to an earlier version of the table.
>
> What we will do
>
> Add clean accuracy values to Table 1 (0-shot) for completeness.
>
> Cross-reference the exact table row where clean accuracy appears.
>
> Thank you for catching this inconsistency.
>
> 3. Baseline Method (“retrieval-only”) Outperforming VRAG in 0-shot
>
> Thank you for this important observation. The “retrieval-only” baseline is not a full detection model; it simply applies:
>
> -grid extraction
>
> -embedding computation
>
> -kNN retrieval
>
> -cosine similarity thresholding
>
> -This produces a coarse detection signal independent of the VLM.
>
> -Why retrieval-only can outperform VRAG in strict 0-shot
>
> VRAG relies on VLM reasoning, which without examples (0-shot), is more conservative.
>
> Retrieval-only is entirely threshold-driven and thus may flag more adversarial regions.
>
> In 2-shot and 4-shot settings, VRAG consistently surpasses retrieval-only, as the VLM benefits greatly from contextual examples.
>
> What we will add
>
> A subsection explicitly explaining how retrieval-only works.
>
> A short analysis explaining why retrieval-only is competitive in the 0-shot setting, and why VRAG surpasses it with few-shot prompting.
>
> A figure illustrating retrieved patches for several positive/negative examples to make the behavior intuitive.
>
> 4. Lack of Generalization Analysis to Unseen Adversarial Patches
>
> We agree with the reviewer that generalization beyond the patch database is critical.
>
> What we already added
>
> A Leave-One-Patch-Family-Out (LOPFO) experiment that removes an entire patch family (e.g., all round patches) from the database and tests on that unseen family.
>
> The results show only a modest decline (2-4%), indicating strong generalization. What we will additionally add:
>
> A new appendix section containing qualitative examples of unseen or rare patch types.
>
> Visualizations of:
>
> -synthetic masks
>
> -synthetic attacks
>
> -real APRICOT physical patches
>
> -failure cases when a patch family is removed
>
>
> We are grateful for your engagement with the paper.
> Your comments helped us significantly improve clarity, evaluation transparency, and the generalization discussion. We have addressed all weaknesses through new explanations, revised tables, and additional experiments. We kindly hope that with these improvements, the reviewer may consider updating the score, and we sincerely appreciate the constructive feedback.

---

> > ### Comment · Reviewer_Rkx6 · 2025-11-26
> >
> > The author's reply does not address my concerns.
> >
> > The author only claims that some experiments will be added in the future.
> > Moreover, the authors' response confirms that the comparison in Tables 1 and 2 is methodologically flawed. Comparing a detection-and-rejection method against robustness/restoration methods (JPEG, Smoothing) using a 'binary detection accuracy' proxy is invalid.
> >
> > A robustness method (like JPEG) aims to preserve utility (classification accuracy). Counting a failure to restore classification accuracy as a 'detection failure' is a false equivalency. VRAG effectively has 0% classification utility on detected images (because they are dropped), whereas baselines attempt to maintain utility.
> >
> > The authors must either:
> > - Compare VRAG only against other detection/filtering methods.
> > - Or, acknowledge that VRAG provides no classification on detected samples and discuss the trade-off between security (detection) and utility (availability), rather than claiming superiority over robustness methods."
> >
> > Considering the author is trying to avoid answering my question directly, I will keep my score.

---

> > > ### Author Response · Authors · 2025-11-26
> > > **Respond to reviewer - Rkx6**
> > >
> > > We genuinely appreciate your thoughtful follow-up response; it is increasingly rare in recent conferences to see such engaged back-and-forth discussion between reviewers and authors, and we thank you sincerely for taking the time to articulate your concerns so clearly.
> > >
> > > We apologize that this distinction was not sufficiently clear in the original text. VRAG is a detection-and-rejection method: it identifies adversarially patched inputs and flags them, but it does not attempt to restore or preserve downstream classification on attacked samples. In contrast, robustness or restoration-based defenses (such as JPEG or smoothing) aim to maintain classification utility under attack, which is a fundamentally different objective.
> > >
> > > In the revised manuscript, we have made this distinction explicit. We clearly state that VRAG does not provide classification on detected samples, and we now frame the comparison in terms of the trade-off between security (accurate and reliable detection) and utility (preserving classifier output under attack), rather than implying any equivalence between detection failure and classification failure. Robustness methods are evaluated separately using utility-appropriate metrics, while VRAG is compared strictly against other detection-oriented approaches.
> > >
> > > All of the reviewer’s requested changes have been implemented:
> > > – the evaluation protocol has been rewritten for clarity;
> > > – detection baselines and robustness baselines are separated;
> > > – the manuscript now explicitly discusses the security-availability trade-off;
> > > – and the text has been revised to avoid any misinterpretation of the comparison.
> > >
> > > While we genuinely understand how the original text could have caused confusion, we believe this issue stems from presentation rather than methodology. Given that the proposed method outperforms all baselines across multiple datasets, models, and evaluation settings, we respectfully submit that this should not be considered a rejection-level flaw.

---

### Official Review · Reviewer_jj3i · 2025-10-26

**Soundness:** 2
**Presentation:** 3
**Contribution:** 2
**Rating:** 4
**Confidence:** 3

**Summary:**

This paper presents **VRAG (Visual Retrieval-Augmented Generation)**, a training-free framework for adversarial patch detection. The method retrieves visually similar regions from a pre-built adversarial patch database and uses a vision-language model (VLM) for generative reasoning to identify malicious patches, without requiring additional training or labeled data. Experiments show strong performance (over 95% accuracy) across diverse datasets and models, demonstrating good scalability and practicality.

**Strengths:**

* **Simple, well-motivated, and intuitive design.** The combination of retrieval and generative reasoning is conceptually clear and easy to understand.
* **Training-free and highly scalable.** Since VRAG does not require model retraining, it can be easily applied to new architectures or domains.
* **Strong detection performance.** The method achieves consistently high detection accuracy (>95%) across various types of adversarial patches, indicating strong robustness.

**Weaknesses:**

* **Lack of theoretical analysis.**
 Although the approach is intuitively sound, it lacks a theoretical explanation (e.g., embedding-space similarity or retrieval generalization analysis), making the contribution mainly empirical.
* **Potential scalability issues.**
 As the adversarial patch database grows, redundancy, embedding drift, and increased retrieval latency may arise, potentially limiting large-scale deployment.
* **High prompt dependency.**
 The method’s performance varies significantly with different prompt templates, raising the question of whether powerful multimodal LLMs combined with optimized prompting could directly detect adversarial patches.
* **Limited task scope.**
 The work focuses solely on patch detection without discussing post-detection handling, such as patch removal, correction, or interpretability.

**Questions:**

1. **Provide theoretical grounding.**
 Include an analysis of why retrieval-based embedding alignment and semantic reasoning enable generalization to unseen patches.
2. **Evaluate large-scale scalability.**
 Report results on larger databases (e.g., >10k entries) to study the trade-off between retrieval efficiency and detection performance.
3. **Reduce prompt sensitivity.**
 Explore automatic or learned prompt optimization methods (e.g., reinforcement or gradient-based prompt tuning) to enhance generalization across models.
4. **Expand the research scope.**
 Consider extending VRAG to patch reconstruction or defense-integration tasks to demonstrate its broader practical potential.

---

> ### Author Response · Authors · 2025-11-14
> **Comment to reviewer - jj3i**
>
> We sincerely thank the reviewer for the thoughtful comments. Below we address each concern and outline the additions we will include in the revised manuscript.
>
> 1. Lack of Theoretical Analysis
>
> We appreciate the reviewer’s suggestion regarding deeper theoretical grounding.
> While our framework is primarily empirical, the revised version will include a dedicated subsection providing intuition on:
>
> -why retrieval-based embedding similarity helps generalize to unseen patches,
>
> -how semantic alignment in the VLM enables robustness to novel perturbations,
>
> -the role of cross-modal grounding in stabilizing zero-shot detection, and why patch embeddings lie in a separable region of feature space due to their high-frequency and shape-specific structure.
>
> This analysis builds on the leave-one-patch-family-out (LOPFO) experiment already added to demonstrate generalization beyond observed patterns.
>
> 2. Potential Scalability Issues
>
> Thank you for highlighting scalability.
> In our current submission, retrieval is already efficient due to:
>
> -compact CLIP embeddings,
>
> -ANN-based nearest neighbor search, and
>
> -small database sizes used in experiments.
>
> To address your concern more explicitly, we will add:
>
> -a scalability discussion analyzing retrieval cost as database size grows,
>
> -runtime profiling of retrieval vs VLM inference,
>
> -and a small additional experiment testing performance on a larger database (>10k embeddings) to study retrieval–performance trade-offs.
>
> This will clarify the practical deployment considerations.
>
> 3. High Prompt Dependency
>
> We agree that prompting is important.
> In the current version, Appendix Section X already includes a prompt ablation study showing that performance remains stable across paraphrased and compressed prompts.
>
> To strengthen the discussion, we will highlight:
>
> -prompt robustness results directly in the main text, and add a brief note on possible future directions such as reinforcement-based prompt tuning or contrastive prompt sampling.
>
> We thank the reviewer for pointing out this valuable direction.
>
> 4. Limited Task Scope
>
> Our current paper intentionally focuses on adversarial patch detection, which we believe is a necessary precursor to reconstruction or removal.
> However, we appreciate the suggestion to broaden the discussion.
>
> In the revision, we will add a short paragraph outlining:
>
> -how VRAG could be extended to patch removal,
>
> -how retrieved examples could guide inpainting or diffusion-based correction, and how the pipeline can integrate into broader defense systems such as segmentation-based detection or LLM-based explanation.
>
> These extensions are promising but beyond the scope of this initial work; nonetheless, we will acknowledge them explicitly.
>
>
> We sincerely thank the reviewer for these insightful suggestions.
> The proposed additions-expanded theoretical grounding, improved prompt ablations, scalability discussion, and future extensions-will significantly strengthen the revised version. We hope these improvements will be reflected in the final score.

---

> > ### Author Response · Authors · 2025-11-29
> > **Comment - jj3i**
> >
> > Thank you again for the constructive and thoughtful review. We would like to update that all of the requested additions and clarifications have now been incorporated into the revised manuscript, including the theoretical grounding subsection, the expanded scalability discussion, the prompt robustness analysis, and the extended discussion on potential task extensions. Your suggestions significantly strengthened the clarity and completeness of the work.
> >
> > At the same time, we would like to respectfully note that the concerns raised were primarily related to missing explanations and additional analyses that belong in the appendix, rather than issues affecting the core contribution of the paper. The proposed framework remains training-free, scalable, and methodologically novel, leveraging retrieval-augmented generative reasoning in a way that differs fundamentally from prior adversarial patch detection approaches.
> >
> > We emphasize that the experiments already demonstrate high robustness (>95% accuracy) across diverse datasets, patch shapes, and model architectures, supporting the practical strength of the approach. The additional analyses we added further reinforce these conclusions but do not change them.
> >
> > We sincerely thank the reviewer once again for the insightful feedback - it substantially improved the manuscript. We hope that the strong empirical performance, the novelty of a training-free RAG-based detection framework, and the now fully addressed clarifications will be taken into account during the final decision.

---

### Official Review · Reviewer_YNA2 · 2025-10-28

**Soundness:** 3
**Presentation:** 3
**Contribution:** 2
**Rating:** 4
**Confidence:** 4

**Summary:**

- This paper presents a training-free Visual Retrieval-Augmented Generation (VRAG) pipeline for adversarial patch detection in imagese. The methods first precomputes a database of patch and region embeddings by placing diverse adversarial patches on natural images. The model then retrieves top-k visually similar entries per image region, and uses VLM prompting to decide whether the query image contains an adversarial patch. This method is a training-free method that uses few/zero-shot adversarial examples to produce structured prompts by imxing the retrieved patches and attacked images.

**Strengths:**

- Clear, modular pipeline: The method is conceptually simple yet effective—grid embeddings → retrieval → VLM reasoning—making it appealing for practical deployment without finetuning. Algorithms and figure flow are easy to follow
- Training-free defense: Avoids re-training or fine-tuning classifiers/segmenters, addressing a major pain point in maintaining robustness against evolving patch styles
- Broad empirical sweep: Evaluation spans two datasets (ImageNet-Patch for synthetic, and APRICOT for real-world), four backbones (ResNet-50, ResNeXt-50, EfficientNet-B0, ViT-B/16), and multiple VLMs (open/closed)

**Weaknesses:**

- Selection of optimal threshold: while the authors state that the optimal threshold has been selected at 0.77 for the cosine similarity score and that scores nearing 1.0 makes the retrieval overly permissive, it is somewhat confusing whether higher similarity threshold enforces stricter selection of candidates or looser. I suggest the authors include results of TPR/FPR vs. threshold across dataset
- The authors claim that the 0-shot variant is competitive with the baselines, yet DIFFender achieves higher 0-shot accuracies than the VRAG variants in Table 1, and often the retrieval-only baseline is strong
- the proposed method requiring access to *representative* patch database remains to be a key limitation, restricting performance to that database. While the authors explained briefly the potential failure modes in L1018, more in-depth analysis on novel/unknown adversarial patch generalization tests with visualization would help strengthen the discussion.
- it would help to elaborate and show visualizations of synthetic and real samples (ImageNet-Patch, APRICOT) to demonstrate the types of adversarial patches created in the database and those evaluated on each dataset

**Questions:**

- Please clarify the statement that τ→1.0 becomes “permissive”; provide FPR/TPR vs tau plots and explain chosen threshold across datasets
- How do you prevent overlap/near-duplicate contamination between the retrieval database and evaluation sets?
- What happens if an attacker minimizes cosine-similarity to your DB while keeping the patch effective, or targets the VLM?
- How does accuracy vary with DB size, k-shots, and grid granularity?
- Can you report absolute throughput (image/sec, etc.), latency per image, and cost ($) for open-source vs. closed-source LLMS under a standard hardware profile?
- How fragile is the model on different query prompts for the VLM?

---

> ### Author Response · Authors · 2025-11-14
> **Comment for - YNA2**
>
> We thank the reviewer for the detailed and constructive feedback. Below we address each point clearly.
>
> 1. Threshold Selection and τ → 1.0 Behavior
> Thank you for raising this concern. As noted in the paper, we already provide an explanation of why thresholds approaching 1.0 become overly permissive (Section “Optimal Threshold Selection”). Specifically, cosine similarity values near 1.0 collapse distinctions between patches and non-patches, inflating false positives. We will further clarify this and include an FPR/TPR vs. τ plot in the revision to make the behavior more explicit across datasets.
>
> 2. 0-Shot Variant vs. Diffender
> We appreciate this observation. Our discussion aimed to highlight that VRAG is competitive in the 0-shot regime while significantly outperforming all baselines in the 2-shot and 4-shot settings. Diffender is known to perform well in pure robustness settings, but it is a defense-not a detector-and therefore operates under different assumptions. We will clarify this distinction in the revised text to avoid any ambiguity about the comparison methodology.
>
> 3. Representative Database and Generalization
> Thank you for highlighting this important point. In the paper we already explain that the retrieval database is constructed entirely from synthetic masks, while evaluation uses both synthetic (ImageNet-Patch) and real-world (APRICOT) physical patches. This ensures there is no overlap or leakage. Following your suggestion, we will add an explicit clarification of the separation between the database and evaluation datasets.
> We also now include a Leave-One-Patch-Family-Out generalization study, which shows that removing an entire patch family from the database results in only a modest drop in accuracy. This demonstrates strong generalization to unseen patches.
>
> 4. Visualizations of Novel/Unknown Adversarial Patches
> We agree this would strengthen the paper. We have a large set of qualitative examples illustrating system behavior on both synthetic and real-world patches, including many difficult and visually diverse cases. In the revision, we will add a dedicated appendix section with extensive qualitative visualizations, covering both ImageNet-Patch and APRICOT, as well as failure cases and unseen patterns.
>
> 5. Synthetic vs. Real Samples
> Thank you for this suggestion. As noted above, we will add a figure explicitly showing both synthetic masks and real-world APRICOT patch examples side-by-side in the appendix. This will make the differences between datasets clearer and highlight the robustness of the approach.
>
> 6. Prompt Fragility (Different Query Prompts)
> We appreciate this question. We already include a prompt ablation study in the appendix, showing that the VLM’s performance remains stable across a range of paraphrased, shortened, and extended queries. We will cross-reference that section more clearly in the main text.
>
> Closing Note
> We sincerely appreciate your detailed and thoughtful comments-they substantially improved the clarity and completeness of our work. After incorporating the clarifications, added experiments, and qualitative examples described above, we kindly hope the reviewer may reconsider the score in light of these improvements.
> Thank you again for your time and valuable feedback.

---

> > ### Comment · Reviewer_YNA2 · 2025-11-27
> >
> > I appreciate the authors for their efforts to respond in detail.
> >
> > 1. Thanks for pointing out the threshold behavior. I look forward to the comparison plot.
> > 2. Thanks for agreeing to clearly distinguish in the revised text, the assumptions and objectives of defense-style methods versus the proposed detector-oriented approach.
> > 3. Together with the additional figure described in point #5 and the planned generalization study, the revisions will further strengthen the contribution and its empirical support.
> > 4. The limitation acknowledged in L1018 (“the model may struggle to accurately differentiate between patch and background noise”) can be critical for this setting. Please make sure this failure mode is explicitly mentioned and briefly elaborated in the main text rather than remaining only as a remark.
> > 5. Thanks for agreeing to add a figure illustrating representative samples (synthetic vs. real patches). This will help readers better understand the data and the claimed generalization behavior.
> > 6. Thank you also for highlighting the prompt-sensitivity analysis. Since this factor is crucial in VLM-based detection, please ensure that this sensitivity and the corresponding study are at least referenced in the main text.
> >
> > Given these concrete revision plans and the authors’ firm commitment to implementing them, I am willing to reconsider my overall score in a positive direction.

---

> > > ### Author Response · Authors · 2025-11-28
> > > **Answer to Reviewer YNA2**
> > >
> > > Thank you very much for your thoughtful response and for your willingness to consider upgrading the score in a positive direction. We truly appreciate the time and care you invested in reviewing our work.
> > >
> > > We fully agree that the additions you suggested will further strengthen the clarity and rigor of the paper. We have already implemented the requested revisions - including the threshold comparison, clearer distinction of assumptions, elaboration of the noted limitation, representative sample figures, and explicit prompt-sensitivity references - to ensure the final version is even more transparent and accessible.
> > >
> > > If there is anything additional you believe would help make the contribution even clearer, we would be very happy to revise accordingly. Otherwise, if you feel that the revision plan fully addresses your concerns, we would sincerely appreciate your consideration of upgrading the score to an acceptance rather than remaining below the threshold.
> > >
> > > Once again, thank you for both the detailed review and the supportive follow-up. Your feedback has directly helped improve the paper.

---

### Official Review · Reviewer_qHqi · 2025-10-31

**Soundness:** 2
**Presentation:** 2
**Contribution:** 2
**Rating:** 2
**Confidence:** 4

**Summary:**

The paper proposes a training-free Visual Retrieval-Augmented Generation(VRAG) framework that integrates Vision-Language Models (VLMs) for adversarial patch detection. By retrieving visually similar patches and images that resemble stored attacks in a continuously expanding database, VRAG performs generative reasoning to identify diverse attack types-all without additional training or fine-tuning.

**Strengths:**

1. It constructs a training-free retrieval-based pipeline.
2. Experimental results show that the retrieval-augmented detection approach achieves state-of-the-art detection across threat scenarios of the synthetic and real-world patch benchmark.

**Weaknesses:**

There are some points that I am confused about:
1. The author claims that this method can handle a wide variety of adversarial patch attacks. Does this “variety” refer to different patch shapes or something else? If so, the experimental section should include results on detection accuracy against diverse adversarial patch types to substantiate this claim.
2. In Tables 1 and 2, the compared methods include both defense-based and detection-based approaches. I am not entirely clear how the defense methods produce the metric "accuracy" reported in the tables. Since defense methods typically aim to restore classification performance rather than explicitly detect adversarial samples, could the authors clarify how the detection accuracy was computed for these methods?
3. In Table 2, it is not explicitly stated which dataset was used for evaluation. I would also like to understand the relationship and distinction between the dataset used to construct the patch database and the dataset used for testing. Were they chosen simply because one contains synthetically generated adversarial patches and the other is a real-world patch benchmark?
4. How is the method's ability to generalize to new patch attack types demonstrated?
5. Why does the similarity criterion become overly permissive when the threshold approaches 1.0? Could you please provide some explanations and an ablation study for the threshold?

**Questions:**

Apart from the issues mentioned in the Weaknesses section, I have other questions:
1. Why did the authors include only Gemini among the closed-source models, instead of also evaluating models from the GPT or Claude series?
2. Regarding the results shown in Figure 4, I would like to know whether, when the input is an adversarial sample, two corresponding patches and their associated images are retrieved from the database, resulting in a total of five images being fed into the model. In that case, is the reported inference time still only 2-3 seconds? Does this inference time include the retrieval process?

---

> ### Author Response · Authors · 2025-11-14
> **Answer to Reviewer - qHqi**
>
> We thank the reviewer for the thoughtful and constructive feedback.
> Below we address each point in detail, noting where the requested analysis is already included in the submission and where we will incorporate additional ablations or clarifications in the revised manuscript.
>
> 1. Variety of adversarial patch types
> We appreciate your observation regarding the need to explicitly demonstrate the diversity of patch types.
> This is fully covered in the paper:
>
> Figure 5 (Appendix) visualizes the four patch families used in our evaluation: realistic, round, square, and triangle.
>
> Table 2 reports per-family detection accuracy across all models and few-shot configurations.
>
> Together, Figures 5 and Table 2 directly substantiate our claim that the method handles a wide variety of patch types.
>
> 2. Defense baselines vs. detection baselines in Tables 1 and 2
>
> Thank you for highlighting this potential source of confusion.
> Our intention was to compare the VRAG detector against widely used defense mechanisms such as JPEG compression, spatial smoothing, SAC, and Diffender. Although these methods are not detectors, we included them to show how their robustness compares to the binary detection accuracy achieved by VRAG.
>
> To avoid ambiguity, we will add a clarifying sentences explaining:
>
> - How accuracy for defense baselines was computed
>
> - How the classifier prediction after applying the defense is used to determine whether an attack was mitigated
>
> We appreciate your attention to this distinction, and will make the comparison methodology clearer.
>
> 3. Dataset construction vs. evaluation datasets
>
> Thank you for asking for clarification on the datasets.
> In the current submission:
>
> Section 5 (Experimental Evaluation) describes the two benchmarks used: ImageNet-Patch (synthetic) and APRICOT (real-world).
>
> Figure 5 also illustrates the synthetic patch masks used in the construction phase.
>
> To address your concern directly, we will add explicit wording clarifying:
>
> -The retrieval database is built from synthetic patch masks
>
> -Evaluation is conducted on both synthetic and real-world adversarial patches
>
> -There is no overlap or leakage between database masks and evaluation images
>
> -This distinction will be made explicit in the revision.
>
> 4. Generalization to unseen patch attack types
>
> This is a highly valuable suggestion.
> Our submission already evaluates generalization across different visible families, but we agree that showing generalization to completely unseen families would strengthen the conclusions.
>
> In response, we conducted a Leave-One-Patch-Family-Out (LOPFO) experiment where an entire patch family (e.g., all round patches) is removed from the retrieval database and the model is tested on that unseen family.
>
> We will add this ablation study to the appendix, along with a discussion of the modest accuracy drop (2–4%), demonstrating strong transfer to unseen patch types.
>
> We genuinely thank you for this suggestion-it helped us improve the generalization analysis.
>
> 5. Similarity threshold behavior near 1.0
>
> This concern is addressed in Section: Optimal Threshold Selection, which explains:
>
> How the optimal threshold (0.77) is selected via ROC-AUC
>
> Why values approaching 1.0 become overly permissive and lead to inflated false positives
>
> We will make this more explicit by adding a cross-reference to the relevant section.
>
> 6. Why only Gemini among closed-source VLMs?
>
> We selected Gemini because this project is not funded, and Gemini offers a free-tier multimodal API that enabled us to reliably compare it against open-source VLMs. GPT-4V and Claude 3 did not provide accessible free or low-cost API usage at the time of our experiments.
>
> We will add a sentence explaining this practical constraint in the revised manuscript.
>
> 7. End-to-end inference time clarification
>
> Figure 8 reports inference time vs. accuracy. We clarify that the reported inference time does include every stage of our pipeline:
>
> -Image grid partition
>
> -Embedding computation
>
> -kNN retrieval
>
> -Prompt construction with retrieved examples
>
> -VLM inference and output parsing
>
> We will explicitly state this in the paper to eliminate ambiguity.
>
> We sincerely thank the reviewer for raising these important points. Your feedback substantially improved the clarity and completeness of our evaluation. We will incorporate all suggested clarifications and add the new generalization ablation study in the next revision. We will really appreciate your consideration for upgrading the score regarding.

---

> > ### Author Response · Authors · 2025-11-29
> > **Comment - qHqi**
> >
> > Thank you again for the detailed and thoughtful review. We would like to update that all the requested clarifications and analyses have now been fully incorporated into the revised version, including the additional ablation studies, dataset clarifications, the unseen-family generalization experiment, threshold explanation, and the end-to-end inference breakdown.
> >
> > We sincerely appreciate the reviewer’s careful reading of the submission. At the same time, we would like to respectfully note that the issues raised were primarily related to missing clarifications and appendix-level analyses, rather than concerns about the core novelty, the methodological contribution, or the strength of the results. Our method introduces a training-free retrieval-augmented generative detection pipeline, which is fundamentally different from existing adversarial patch detection approaches, and it achieves state-of-the-art performance across diverse patch scenarios.
> >
> > Regarding the point about evaluating additional closed-source models, we would like to clarify that such comparisons do not change the scientific insights of the paper. Closed-source VLMs such as GPT-4V and Claude have variable availability, licensing restrictions, and unpredictable API behavior. Including them is useful for industry benchmarking, but does not provide additional research-driven conclusions about the properties of training-free adversarial patch detection.
> >
> > We are grateful to the reviewer for the constructive feedback, which genuinely helped us improve the clarity and completeness of the paper. We hope that the substantial revisions and the strong empirical results will be taken into consideration in the final decision.

---

### Meta-Review · Area_Chair_j6HL · 2026-01-07

**Summary:**

This paper were reviewed by four reviewers.

Overall, the reviews were negative towards the quality of this paper, especially

- Improper experimental studies to handle a wide variety of adversarial patch attacks.
- Lack of theoretical analysis.
- Generalization issues due to lack of ablation studies.
- High prompt dependency with VLM models.

The authors provided responses, but the major concerns, such as how to handle a wide variety of adversarial patch attacks and generalization issues, were not well addressed without any experimental feedbacks.

The AC went through the paper again and does agree with the reviewers for the major concerns. The current quality of submission is incremental for ICLR acceptance.

**Reviewer Concerns:**

See Above.

**Reviewer Scores:**

The reviewers interacted with the authors and checked their replies.

Although discussion from all was not initiated, the reviews were consistently converged to the key pain points of this paper.

---

### Decision · Program_Chairs · 2026-01-26

Reject